# Estimating quality-adjusted life expectancy (QALE) for local authorities in Great Britain and its association with indicators of the inclusive economy: a cross-sectional study

Andreas Höhn [1], Nik Lomax [2], Hugh Rice [2], Colin Angus [3], Alan Brennan [3], Denise Brown [1], Anne Cunningham [3], Corinna Elsenbroich [1], Ceri Hughes [4], Srinivasa Vittal Katikireddi [1], Gerry McCartney [5], Rosie Seaman [1], Aki Tsuchia [3], Petra Meier [1]

For numbered affiliations see end of article.

**Correspondence to**
Andreas Höhn;
andreas.hoehn@glasgow.ac.uk

## ABSTRACT

**Objectives** Quantifying area-level inequalities in population health can help to inform policy responses. We describe an approach for estimating quality-adjusted life expectancy (QALE), a comprehensive health expectancy measure, for local authorities (LAs) in Great Britain (GB). To identify potential factors accounting for LA-level QALE inequalities, we examined the association between inclusive economy indicators and QALE.

**Setting** 361/363 LAs in GB (lower tier/district level) within the period 2018–2020.

**Data and methods** We estimated life tables for LAs using official statistics and utility scores from an area-level linkage of the Understanding Society survey. Using the Sullivan method, we estimated QALE at birth in years with corresponding 80% CIs. To examine the association between inclusive economy indicators and QALE, we used an open access data set operationalising the inclusive economy, created by the System Science in Public Health and Health Economics Research consortium.

**Results** Population-weighted QALE estimates across LAs in GB were lowest in Scotland (females/males: 65.1 years/64.9 years) and Wales (65.0 years/65.2 years), while they were highest in England (67.5 years/67.6 years). The range across LAs for females was from 56.3 years (80% CI 45.6 to 67.1) in Mansfield to 77.7 years (80% CI 65.11 to 90.2) in Runnymede. QALE for males ranged from 57.5 years (80% CI 40.2 to 74.7) in Merthyr Tydfil to 77.2 years (80% CI 65.4 to 89.1) in Runnymede. Indicators of the inclusive economy accounted for more than half of the variation in QALE at the LA level (adjusted $R^2$ females/males: 50%/57%). Although more inclusivity was generally associated with higher levels of QALE at the LA level, this association was not consistent across all 13 inclusive economy indicators.

**Conclusions** QALE can be estimated for LAs in GB, enabling further research into area-level health inequalities. The associations we identified between inclusive economy indicators and QALE highlight potential policy priorities for improving population health and reducing health inequalities.

## STRENGTHS AND LIMITATIONS OF THIS STUDY

⇒ We used a small-area approach to model mortality rates and health state utility scores—the two components required to estimate quality-adjusted life expectancy (QALE).

⇒ The small-area approach described is suitable for estimating other health and state expectancy measures and is easily transferable to other national contexts for which similar data are available.

⇒ As with most measures of health expectancy for small areas, QALE point estimates come with some degree of uncertainty, primarily due to their reliance on survey data.

⇒ Differences in sampling strategies and non-response bias between local authorities in the Understanding Society survey might have contributed to differences in health state utility scores between areas.

## INTRODUCTION

Period life expectancy (LE) is a summary measure of age-specific mortality rates, reflecting a particular point in time.[1] A limitation of LE is that it does not include any assessment of health. Health expectancy measures seek to address this limitation and capture both length of life and health status.

A variety of health expectancy measures are available for research and public health planning, including healthy LE (HLE), disability-free LE (DFLE)—and more recently quality-adjusted LE (QALE). All three measures of health expectancy use life tables to quantify the length of life.[2 3] However, these three health expectancy measures differ substantially in how they measure the health status of individuals. HLE and DFLE measure health in binary terms, by capturing individual-level health at every age as a value

of either 1 or 0. HLE and DFLE reflect a single dimension of health (either 'healthy' vs 'unhealthy' or 'without disability' vs 'with disability'). QALE captures health in far greater detail using individual-level health state utility scores (hereafter: utility scores).

Individual-level utility scores measure health at every age as a value between 0 (dead) and 1 (perfect health).[3] Typically, scores are estimated from standardised questionnaires such as the EuroQol 5-dimension (EQ-5D) or the Short Form 12 (SF-12) tools—all of which capture a range of health dimensions.[4 5] Due to this greater detail, QALE captures both mental and physical health and is more sensitive to changes in health that would typically be masked by the binary definitions used for HLE or DFLE. This places QALE conceptually close to the gold standard measure, quality-adjusted life-years (QALYs), which is the standard metric used in health economics to evaluate cost-effectiveness.[6] While QALYs are typically referred to as an outcome measure in longitudinal study designs, QALE reflects the application of the underlying concept of utility scores to a synthetic life table cohort—analogously to the distinction enabled by other measures of health expectancy (eg, disability-free life-years vs DFLE).

A growing number of studies have used QALE and QALYs to estimate the impact of shocks and interventions on population health. For example, the concepts have been used as an outcome measure to understand the health burden of COVID-19[7] and to shed light on the health impact of diseases[8 9] or adverse life course events.[10] The concepts are also used in simulation studies to evaluate health technologies and policies.[11] Despite the concept's increasing popularity, QALE estimates are often not available for small areas. While QALE has been estimated for the general English population,[12] and broken down by deprivation quintile,[13] no QALE estimates are available for other Great Britain (GB) nations or GB local authorities (LAs). In this paper, we present a small-area approach for estimating QALE for LAs (lower tier/district level) in England, Scotland and Wales.

Despite continued economic growth, GB has experienced a stalling of improvements in mortality and health, alongside widening area-level inequalities.[14] In response, creating a more inclusive economy has increasingly gained traction as a suitable framework to tackle health inequalities. In an exploratory analysis, we examined potential factors accounting for inequalities in QALE at the LA level using 13 inclusive economy indicators, operationalised by the System Science in Public Health and Health Economics Research (SIPHER) consortium. SIPHER is a multidisciplinary group of scientists and government partners at local, regional and national levels.[15] Focusing on a range of different domains, the concept of the inclusive economy approaches the extent and nature of economic inclusion and participation. As economies are rebuilt following the COVID-19 pandemic, the concept of the inclusive economy might be more important for explaining population health outcomes and health inequalities than measures of economic growth or output.[16]

## Estimating life tables

QALE requires information on two components: mortality and health. To derive the mortality component, we estimated period life tables based on mid-year population estimates and death counts. Mid-year population estimates for all three GB nations, as well as death counts for England and Wales, were accessed via the Office for National Statistics' (ONS) web platform NOMIS. We accessed NOMIS via its underlying application programming interface (API). Death counts for Scottish LAs in the respective period were obtained from the website of National Records of Scotland. For both the mortality and health components of QALE, we consistently applied the ONS' definition of LA boundaries as of April 2021. We pooled data for the years 2018, 2019 and 2020.

Despite pooling data for three consecutive years, estimating robust age-specific mortality rates for an area with a small population size is challenging due to small and fluctuating numbers.[17 18] The potential imprecision in age-specific mortality rates increases as the size of the area decreases and fewer deaths are recorded.[19] One approach to address this challenge is to apply specific small-area estimation techniques, such as the tool for projecting age-specific rates using linear splines (TOPALS) models.

TOPALS models have previously been used to estimate mortality rates for small areas.[20–23] Given the general shape of a reference age pattern, TOPALS models estimate age-specific mortality rates by using penalised iteratively reweighted least squares. Reference age patterns should allow for smooth patterns[17] and align to the respective application.[23] We used data from the Human Mortality Database for the general populations of England and Wales, and Scotland in the period 2010–2019 as a reference mortality schedule.[24] We corrected for potential overestimation or under-estimation of mortality after age 90[25]—a phenomenon that can occur due to the functional form of TOPALS models.[23]

We followed standard methodology to derive life tables.[26] An example of sparse mortality data for LAs with small population sizes (eg, males from the Shetland Islands) is presented in figure 1. Figure 1 shows raw death counts and raw death rates for 5-year age groups—as well as the modelled mortality rates for 1-year age groups.

## Estimating utility scores

To derive the health component of QALE, we estimated age-specific and sex-specific utility scores using a special licence area linkage of the Understanding Society survey. Expressed on a scale between 0 (dead) and 1 (perfect health), utility scores summarise an individual's health by combining information on physical and mental health dimensions in one number.[3 27] Understanding Society is a large-scale, randomly sampled, panel survey which is representative of the UK population.[28] A special licence was required to identify the LA of each interviewed

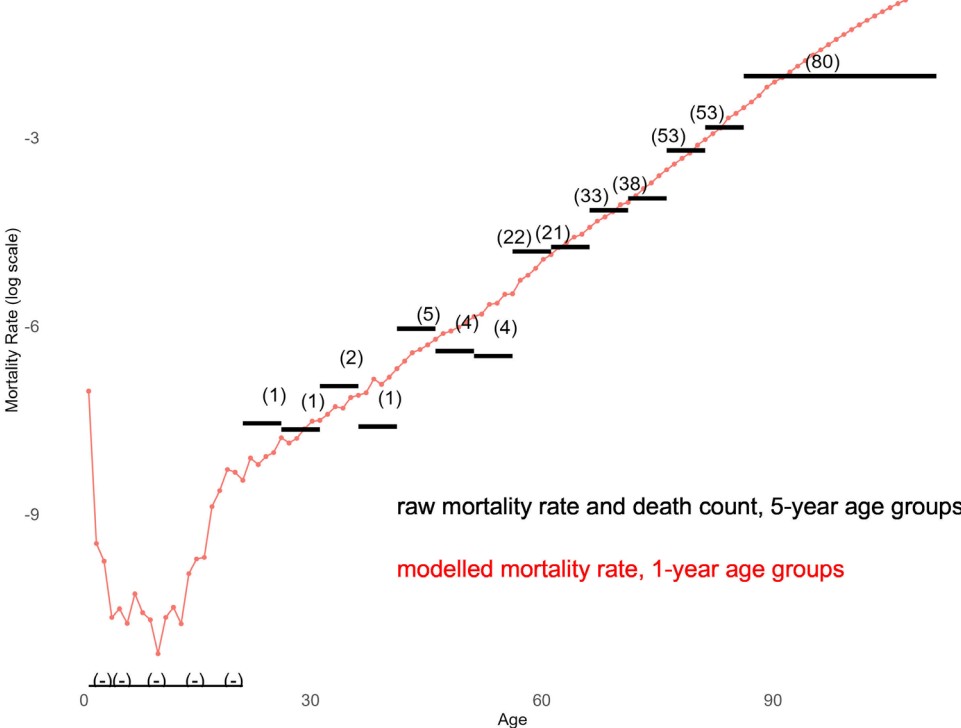

**Figure 1** Number of observed deaths, the resulting raw death rate (log scale) and estimated mortality rates for males (log scale) of the Scottish local authority Shetland Islands, in 2018–2020.

household. We used wave 10 ('j'), which covered the period 2018–2020, the same period for which we estimated life tables.

In Understanding Society, the SF-12 (V.2) measure is used to capture the health of individuals. For each survey participant in Understanding Society, the SF-12 provides a summary score for the physical (PCS) and mental health (MCS) status. In wave 10 ('j'), SF-12 data were available for 16 481 females and 13 118 males.

Using cross-sectional survey weights, we calculated raw age-specific mean PCS scores and MCS scores, separately for males and females of each LA. However, as these raw age-specific and sex-specific mean PCS scores and MCS scores were generally noisy and not always available for all ages among some LAs with very low numbers of respondents, we applied a modelling approach.

Using two weighted linear regression models, we predicted age-specific mean SF-12 PCS scores and MCS scores for males and females in each LA from previously calculated raw age-specific and sex-specific mean PCS scores and MCS scores. Both models included covariates for: sex (female/male), age (categorical 10-year age groups allowing for non-linear age trends), an interaction effect between sex and age, and a fixed LA effect (for each LA). We chose this functional form as it balanced statistical fit and overfitting. All males and females aged 19 or younger were combined in the age group 0–19. While the number of respondents varied across LAs, sample sizes were always sufficient for a fixed LA effect to be estimated.

We converted all modelled age-specific and sex-specific mean SF-12 PCS scores and MCS scores into age-specific

and sex-specific mean utility scores. For this purpose, we used the six-variable predictive model outlined by Lawrence and Fleishman[29] to map SF-12 PCS scores and MCS scores to utility scores. Conversions of SF-12 PCS scores and MCS scores into utility scores, via mapping, are common in health economics research with previous studies showing that mapping approaches generally perform well.[30] Figure 2 shows the age-specific and sex-specific utility scores we estimated, and highlights examples for some of the largest LAs in GB and where authors were able to sense-check results with SIPHER's local and national policy partners.

### Quantifying uncertainty of QALE estimates

Our LA-level QALE estimates are subject to some degree of uncertainty. To quantify the magnitude of this uncertainty, we have estimated CIs for all point estimates.

Approached via the Sullivanmethod, the uncertainty of health expectancies arises from the sum of its two component parts: the health and the mortality components.[31] We estimated the uncertainty of QALE point estimates considering these two components and as described by Jagger et al.[31] While the variation of the health component reflects the uncertainty of the utility scores due to the number of survey participants, the variation of the mortality component reflects the uncertainty of estimated death rates and resulting death counts in the population which we obtained via TOPALS models. Across all LAs, the number of survey participants was substantially lower when compared with the actual population size of the respective LA. As a result,

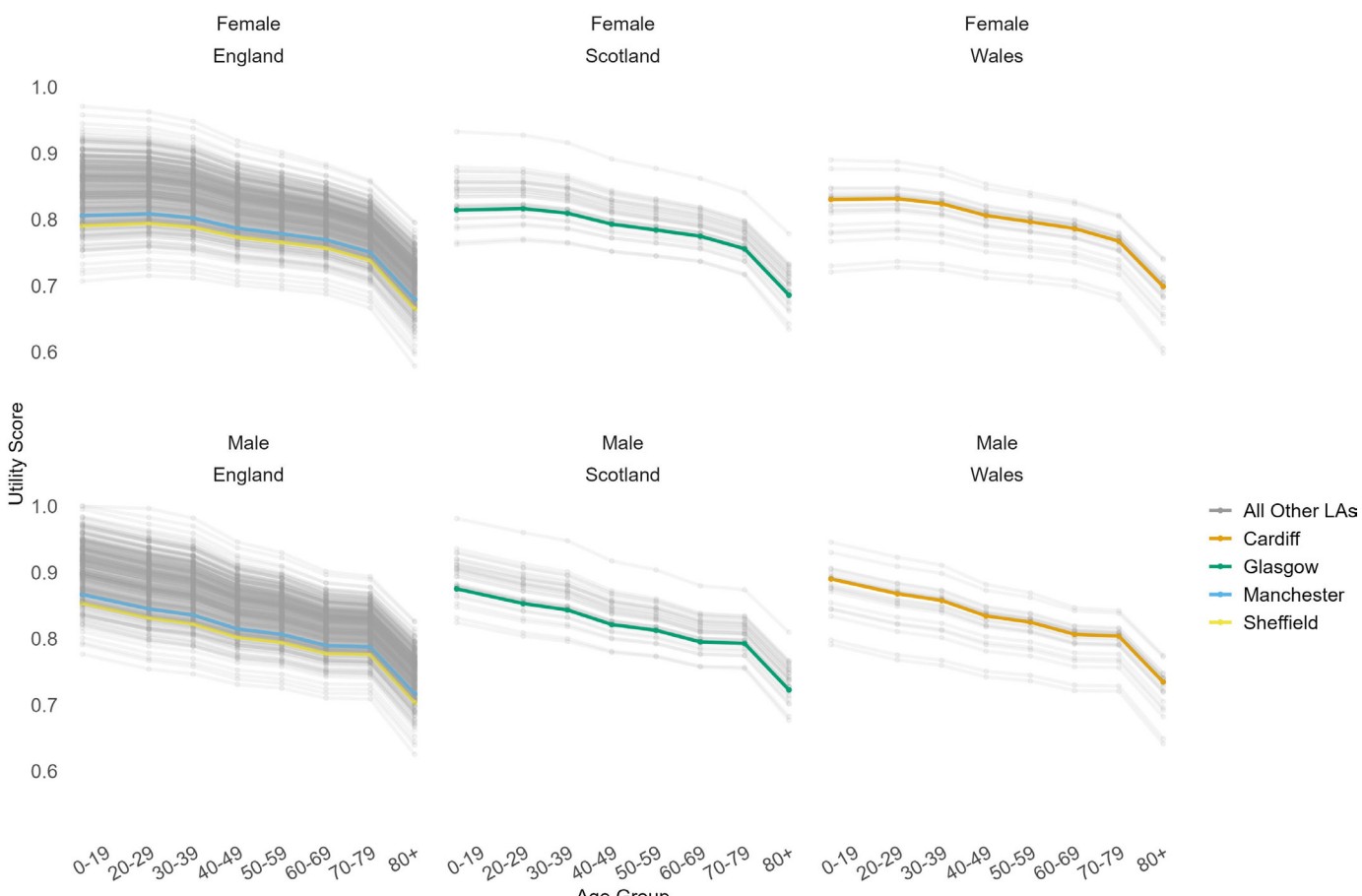

**Figure 2** Utility scores obtained from SF-12 V.2 PCS and MCS mean values for females and males in all local authorities in England, Scotland and Wales with selected local authorities highlighted. LAs, local authorities; MCS, Mental Component Summary; PCS, Physical Component Summary; SF-12, Short Form 12.

the overall amount of uncertainty is clearly dominated by the amount of uncertainty arising from the health component.

We provide 80% CIs, rather than 95% CIs for our point estimates. 80% CIs lead to smaller ranges and reflect 1.282 times the SD as opposed 1.960 times the SD. Using 80% CIs represent a compromise between uncertainty and usefulness with respect to our LA-level QALE estimates. 80% CIs have previously been used for health expectancy estimates for national, regional and super-regional levels.[32]

### Combining life tables and utility scores to estimate QALE

We estimated QALE at birth in years for males and females for each LA using the Sullivan method.[33] The Sullivan method represents an adjustment of the life table parameter nLx, the number of person-years lived between age x and x+1, by the utility score of the age group x. Due to a very small number of respondents in the Understanding Society survey from the LAs City of London and Isles of Scilly, no meaningful QALE estimates could be derived for these LAs. We excluded these areas from all analyses. This reduced the number of LAs for which QALE was estimated to 361 out of 363 LAs in GB.

### Indicators of the inclusive economy

To identify potential factors accounting for inequalities in QALE, we examined the association between indicators of the inclusive economy and QALE. For this purpose, we merged the obtained area-level QALE point estimates with the 13 inclusive economy indicators included in the SIPHER Inclusive Economy (LA level) data set—an open access data set developed by the SIPHER consortium.[34]

To date, there is no single definition of the inclusive economy, making the concept challenging to capture. In response, SIPHER has created an inclusive economy indicator set for all 363 LAs in GB to provide a meaningful collection of data that can be used to explore the extent and nature of economic inclusion. Details of the iterative and stakeholder process for selecting final indicators have previously been described in a technical report.[35] From the inclusive economy (LA level) data set, we used data for the year 2019, the midpoint of our 3-year study period. A descriptive overview of all 13 indicators used in this study is presented in table 1.

We examined the association between indicators of the inclusive economy and QALE using multiple linear regression models. To ensure comparability across all

**Table 1** Descriptive overview of quality-adjusted life expectancy (QALE) at birth, life expectancy (LE) at birth and inclusive economy indicators for all local authorities (LAs) in England, Scotland and Wales studied (n=361 LAs, meaning it does not consider the LAs City of London and Isles of Scilly)

| Indicator | Mean | SD | Min | Max | Explanation |
|---|---|---|---|---|---|
| LE male | 79.68 | 1.92 | 73.37 | 84.19 | Life expectancy |
| QALE male | 67.88 | 3.83 | 57.46 | 77.23 | Quality-adjusted life expectancy |
| LE female | 83.42 | 1.62 | 78.42 | 86.76 | Life expectancy |
| QALE female | 67.70 | 3.84 | 56.34 | 77.72 | Quality-adjusted life expectancy |
| Participation in paid employment | 76.67 | 5.09 | 61.80 | 89.10 | Percentage of working-age population (age 16–64) who are employed. |
| Skills and qualifications | 73.89 | 9.54 | 37.65 | 93.92 | Percentage of adults aged 16–49 with a level 2 or higher National Vocational Qualification. |
| Involuntary exclusion from the labour market | 4.97 | 2.15 | 1.05 | 11.74 | Percentage of working-age population (age 16–64) who are economically inactive due to ill health or disability. |
| Digital exclusion | 33.86 | 20.22 | 0.00 | 87.23 | Percentage of individuals who are classified as e-withdrawn, passive and uncommitted internet users, or settled offline communities, based on the internet user classification. |
| Wealth inequality | 2.68 | 1.06 | 1.31 | 9.30 | Ratio of max to min house sale prices within each local authority. |
| Physical connectivity | 62.50 | 28.02 | 0.00 | 100.00 | Proportion of LSOAs/DZs within the local authority area that are among the 50% most accessible LSOAs/DZs for each devolved nation (56%/44% split in England). |
| Earnings inequality | 3.13 | 0.39 | 2.29 | 5.51 | Ratio of weekly earnings (residents in full-time work) between 80th and 20th percentiles. |
| Housing affordability | 9.92 | 3.62 | 3.95 | 34.46 | Ratio of median house prices to median gross annual earnings (for residents) within each local authority. |
| Poverty | 27.54 | 7.07 | 12.50 | 55.41 | Percentage of children living in low-income households (after adjustment for housing costs). |
| Cost of living | 10.83 | 2.59 | 5.56 | 21.63 | Percentage of individuals reporting that they are worried about getting the food they need. |
| Decent pay | 21.77 | 6.06 | 7.50 | 40.40 | Percentage of employee jobs that are not paid at or above the Living Wage. |
| Inclusion in decision-making | 35.18 | 6.19 | 21.60 | 64.47 | Percentage of individuals participating in local elections out of all eligible voters. |
| Job security | 95.31 | 1.69 | 86.13 | 98.33 | Percentage of individuals in permanent employment out of all employed individuals aged 16–64. |

Values in this table provided for the mean are representative of raw, unweighted averages across all studied LAs. In addition, please note the direction of indicators. For example, a higher employment rate is considered favourable, while a higher proportion of child poverty is not favourable. As some of the indicators reflect ratios, we did not change the direction of indicators. We used the direction provided in the underlying inclusive economy indicator set.

indicators, we used a z-transformation to standardise regression coefficients.

All analyses were performed in R (V.4.3.2). NOMIS API queries were carried out using the NOMISR package.[36] TOPALS models followed the approach and software provided by Schmertmann and Rau.[23 37]

### Patient and public involvement

We did not engage with patients and the public to comment on study design, define outcomes, conduct analyses, interpret results or write the manuscript. However, coproduction within the consortium, involving SIPHER's policy and practice partners, has contributed to different elements of our work on identifying and discussing indicators of the inclusive economy and measures of population health. Results presented in this paper were presented and discussed at workshops with SIPHER's academic and

policy partners. We will continue to disseminate results to our partners at local, regional and national levels.

## RESULTS

### National differences in QALE

QALE differed between GB nations when comparing population-weighted averages across LAs. QALE was lowest in Scotland at levels of 65.1 years among females and 64.9 years among males. Levels were slightly higher in Wales with 65.0 years among females and 65.2 years among males. QALE was highest in England at levels of 67.5 years among females and 67.6 years among males.

### Differences in QALE between LAs in GB

Visualising urban and rural areas with different population densities is challenging. Topographical maps can be

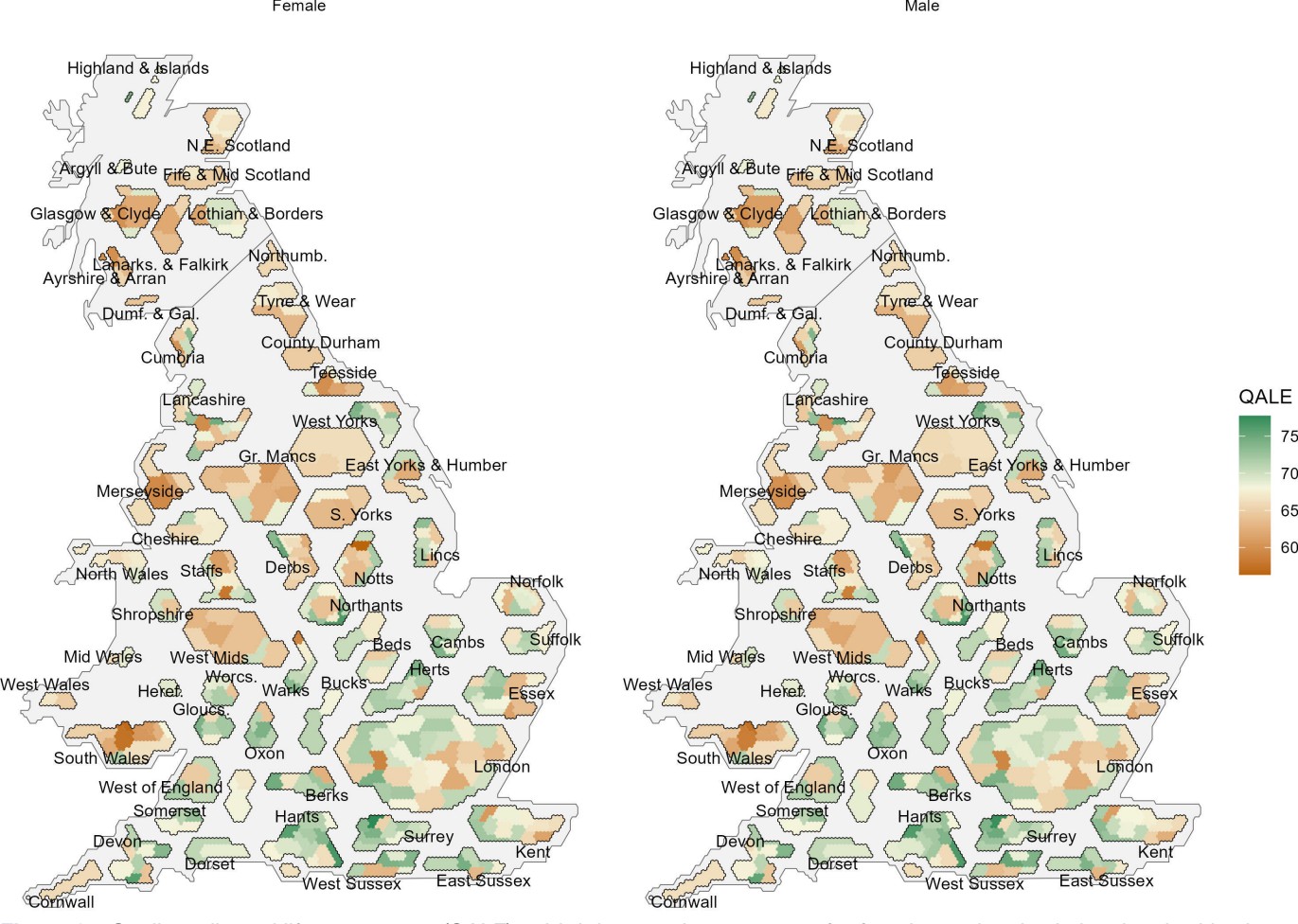

**Figure 3** Quality-adjusted life expectancy (QALE) at birth in years in 2018–2020 for females and males in local authorities in England, Scotland and Wales. The midpoint of the colour scale is referring to the unweighted mean in QALE across all GB LAs. GB, Great Britain; LAs, local authorities.

dominated by large rural areas, overshadowing small but densely populated areas and clustering into higher-level geographies. We used hexagon cartograms to visualise the area-level distribution of QALE for females and males separately (figure 3).[38] This approach visually accounts for the population size of areas, while retaining their contiguity and allowing for clustering into higher-level geographies. A detailed topographical map is presented in online supplemental figure 1.

For females in Scotland, we found QALE to be lowest in North Ayrshire at a level of 60.0 years (80% CI 51.8 to 68.2) and highest in Na h-Eileanan Siar at a level of 74.1 years (80% CI 53.5 to 94.7). Among males in Scotland, QALE was lowest in Renfrewshire at 59.6 years (80% CI 51.7 to 67.5), while it was highest in Na h-Eileanan Siar at 72.7 years (80% CI 51.1 to 94.4). This resulted in a difference of 13.1 years (among females) and 16.1 years (among males) in QALE point estimates across the Scottish LAs with the highest and the lowest level of QALE.

For females and males in Wales, we found QALE to be lowest in Merthyr Tydfil at levels of 56.9 years (805 CI 41.4 to 72.3) and 57.5 years (80% CI 40.2 to 74.7),

while QALE was highest in Vale of Glamorgan at levels of 71.4 years (80% CI 64.6 to 78.3) and 71.4 years (80% CI 65.4 to 77.3), respectively. The corresponding difference in QALE point estimates between Welsh LAs with the highest and the lowest QALE was 14.5 years among females and 13.9 years among males.

Among females and males in England, QALE was lowest in Mansfield at levels of 56.3 years (80% CI 45.6 to 67.1) and 57.7 years (80% CI 48.2 to 67.2), while it was highest in Runnymede at levels of 77.7 years (80% CI 65.1 to 90.3) and 77.2 years (80% CI 65.4 to 89.1). Differences between the LAs with the highest and lowest QALE point estimates were the most striking in England at levels of 21.4 years among females and 19.5 years among males.

### Sex differences in QALE within LAs
When looking at differences between females and males within the same LA, there was always a clear female advantage in LE. However, there was not always a clear female advantage in QALE. For example, females in Glasgow had a 5-year higher LE than males. The corresponding female QALE advantage was less than 1 year in Glasgow: 62.0 years (80% CI 57.4 to 66.7) among females and 61.2

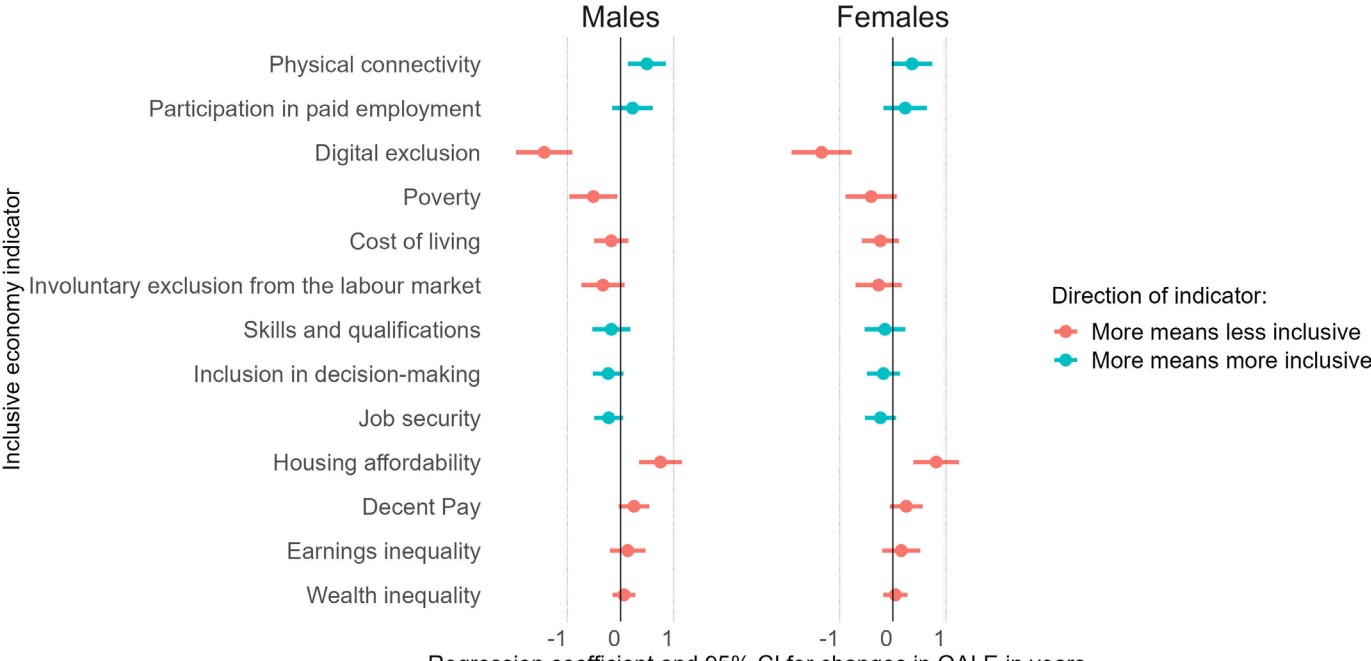

**Figure 4** Multiple linear regression model for the association between inclusive economy indicators and quality-adjusted life expectancy (QALE) at birth in years and respective 95% CIs. Regression coefficients were z-transformed. This means that regression coefficients show the expected change in QALE in years for a 1 SD increase among the indicators, with respect to the GB mean level. GB, Great Britain.

years (80% CI 57.1 to 65.2) among males. For Sheffield, we found a female advantage in LE of approximately 4 years. However, there was a small disadvantage in QALE for females (63.4 years (80% CI 60.3 to 66.4)) when compared with males (63.6 years (80% CI 60.8 to 66.4)).

### Associations between inclusive economy indicators and QALE

We examined the associations between the inclusive economy indicators and QALE to gain insights into potential factors contributing to QALE inequalities at the LA-level in GB (figure 4). Overall, we found that inclusive economy indicators explained more than half of the variation in QALE between LAs in GB (adjusted $R^2$ females/males: 50%/57%). We hypothesised that indicators reflecting more inclusivity would be associated with higher levels of QALE and that indicators reflecting lower levels of inclusivity would be associated with lower levels of QALE. As shown in figure 4, our results were mixed as the direction of associations and levels of significance were not always consistent with our hypotheses.

For 6 out of the 13 inclusive economy indicators, we found the direction of the association between indicator and QALE to be in line with our hypotheses (more inclusive: higher QALE). For example, in relation to the observed mean level of public physical connectivity for the whole of GB (62.50%), an increase by 1 SD (28.02%) was associated with an increase in QALE among females by 0.36 years (95% CI −0.02 to 0.74) and among males by 0.49 years (95% CI 0.14 to 0.85). Consistent with our hypotheses (less inclusive: lower QALE), we also found that an increase in digital exclusion by 1 SD (20.22%), in relation to the observed GB mean level (33.86%), was

associated with a decrease in QALE among females by 1.34 years (95% CI −1.91 to −0.78) and among males by 1.44 years (95% CI −1.97 to −0.91).

For the remaining 7 out of the 13 inclusive economy indicators, the direction of the association between indicator and QALE was not in line with what we hypothesised. For example, we found that an increase in job security was associated with a decrease in QALE among females (−0.23 years; 95% CI −0.53 to 0.06) and among males (−0.23 years; 95% CI −0.50 to 0.05).

One potential mechanism behind unexpected patterns could be residual confounding. For example, with respect to job security, non-permanent employment could be associated with well-paid occupations. High ratios of house prices to earnings (ie, the metric used for the housing affordability indicator) might indeed capture less inclusivity, but could be indicative of affluent suburban areas.

## DISCUSSION
### Principal findings
This paper details an approach for estimating QALE for LAs in GB, enabling further research into area-level health inequalities. Our approach harnesses the power of existing data and is transferable to other national contexts for which required data on mortality and health are available. The results of our exploratory analysis highlight the importance of economic inclusion as a potential policy priority for improving population health and reducing population health inequalities in GB.

## Strengths and limitations

This is the first study to estimate QALE for all LAs in GB. Both the mortality and health components of QALE were derived via statistical modelling techniques, rather than being representative of raw input data only. Therefore, we consider our LA-level QALE estimates to be reliable, and our approach to be transferable to other national contexts for which respective small-area data are available.

As with LE, and most measures of health expectancy such as HLE and DFLE, the estimation of QALE is based on a synthetic cohort approach. This means that our estimates provide timely cross-sectional snapshots—but should not be considered as precise forecasts of QALYs that individuals are likely to experience. In the absence of shocks to population health, such as famines, epidemics and wars, measures of life (and health) expectancy tend to underestimate the years of life (and years in good health) as societal and medical progress are not accounted for.[39] On another note, recent period effects such as the consequences of austerity measures,[40] the COVID-19 pandemic,[41] and the subsequent cost-of-living crisis[42] highlight that deteriorations in population health can occur throughout the life course of individuals.

The sensitivity of life and health expectancy to cross-sectional patterns of mortality and health means that our QALE estimates might have been impacted by the COVID-19 pandemic. The pandemic cannot have had an impact on the health component of our QALE estimates as the collection of survey data occurred throughout 2018 and 2019. However, the pandemic might have had an impact on the mortality component, which covers the years 2018–2020. As a sensitivity analysis, we therefore re-ran our analysis not considering mortality observed throughout the year 2020. A comparison of resulting differences in QALE is presented in online supplemental figure 2. As expected, we found that excluding mortality of 2020 resulted in an (unweighted) increase of QALE of about 0.17 years among females and 0.28 years among males across all studied LAs.

We obtained QALE estimates based on the Sullivan method. As a prevalence-based approach, the Sullivan method stipulates one universal mortality pattern, via the $nLx$ parameter, for all individuals represented by a life table. This is a major methodological drawback, as it is likely that mortality will differ across individuals of different health status.[43]

As with all measures of health expectancy, especially when estimated for small areas, our QALE estimates are subject to a substantial level of uncertainty. This uncertainty is typically much larger than the uncertainty surrounding corresponding LE estimates.[44] As a result, care is required when comparing health expectancy point estimates in longitudinal analyses, in particular across areas with small population sizes. The uncertainty of our QALE estimates arises from the relatively small number of survey respondents contributing to the estimation of the health component. We have decided to not further increase the survey sample size synthetically when estimating the health component of QALE. For example, the survey sample size could have been synthetically increased to reflect the entire adult population in GB using a full-scale synthetic population dataset that we have previously developed, and which is based on the Understanding Society survey.[45] Instead, we have relied on the original sample size of the survey, reflecting its special licence area-level linkage. While the Understanding Society survey is representative of the UK population,[28] sampling strategies and non-response will likely have varied across LAs, deprivation groups and by health status. Weighting and oversampling might have not fully corrected for these limitations.

To estimate life tables, we used official LA-level mid-year population estimates which are rolled forward from the preceding census, using a cohort component model. The estimates based on the 2011 census have recently been revised to align with the 2021 census in England and Wales, but an equivalent adjustment has yet to be made in Scotland. Irrespective of adjustments, unrecorded international migration and migration between LAs could have had an impact on the quality of mid-year population estimates, affecting the estimation of age-specific and sex-specific mortality rates. As the civil registration system in GB does not allow us to follow individuals' places of residence between censuses, this numerator-denominator issue, alongside differences in data availability for the three GB nations, is likely to remain an unresolved challenge for periods between censuses.[46]

We accessed data on mid-year population estimates and death counts for 5-year age groups via NOMIS. These routinely provided data included rounded mid-year population estimates (to 100s), while death counts below 5 were disclosed for LAs in England and Wales. NOMIS data can differ in comparison to bespoke data access requests available via the main ONS website. While bespoke data requests may contain more detailed information, they are not routinely updated—unless rerequested. To maximise reproducibility, transferability and updateability of the underlying pipeline, we queried data in code-based format via NOMIS and its API. This reflects a sustainable and continuously updated source of data. Due to the use of TOPALS models for modelling mortality rates against the general population standard, the overall impact of differences across different data sources is likely to be negligible

## Implications of findings and relevance to other studies

When aggregated and weighted by the population of each LA, our QALE estimates for England are slightly lower than previously reported QALE estimates for the English population in 2017–2018 (females: 67.5 years vs 68.2 years; males: 67.6 years vs 68.2 years).[12] Small differences in values were expected and could be due to a multitude of aspects: differences in the sampling biases in the survey data used, the measure used to capture health (EQ-5D vs SF-12v2), or the exact years reflected. At the same time,

this difference could reflect the observed stall in population health improvements across GB.[40]

Findings of our analysis indicate large inequalities in QALE across LAs in GB, further evidencing substantial area-level inequalities in population health. Our findings for QALE show several similarities when compared with patterns reported for HLE across LAs in GB reported for HLE across LAs in GB.[47 48] For example, in line with results for HLE, the difference in QALE between areas with the highest and the lowest levels was much larger than the difference with respect to LE. This is further evidence that capturing mortality in isolation is likely to underestimate the differences in the burden of ill health experienced across GB.

Generally, we found areas with low levels of QALE to also be among the lowest with respect to HLE. For example, we estimated QALE to be the lowest among males in Merthyr Tydfil at levels of 57.5 years while HLE in the respective period was also low at a level of 58.9 years.[47] Nevertheless, differences in how health is captured means that absolute levels and rankings will likely vary across different measures of health expectancy. At the same time, researchers and policy-makers would benefit from developing a stronger comparative approach which captures the performance of areas with respect to different population health metrics.[49]

When looking within the same LA, females always had higher levels of LE than males. In line with patterns for other health expectancy measures[50] and previously reported results for QALE across population subgroups in England,[51] we found the female advantage in QALE to be very small or negligible. This represents a health-survival paradox: Females live longer but do not necessarily have better health in comparison to males. It is still not fully understood whether this health-survival paradox is the reflection of a true female health disadvantage (despite having a survival advantage), or the result of gender differences in self-reporting health.[52] It is possible that the health component of QALE could be affected by sex-specific survey response behaviour. Sex differences in QALE could stem from sex differences in the SF-12 health instrument which we used to derive utility scores. The SF-12 health instrument captures mental health explicitly. Averaged across LAs, we found the female disadvantage in the SF-12 mental health score (females: 46.8 vs males: 49.2, p<0.01) to be larger than the female disadvantage in the SF-12 physical health score (females: 48.7 vs males: 49.7, p<0.01).

Historically, improvements in population health were often attributed to economic growth.[53] However, this assumption is increasingly contested for high-income countries, including GB, as the association between economic growth and improving health does not seem to hold past certain levels of development.[14 40] In response, there have been growing calls to shift policy ambitions away from economic growth, towards inclusive economies, with a view to achieving more sustainable and equitable health improvements.[16] We found that inclusive economy indicators explained more than half of the variation in QALE between LAs in GB. This highlights the importance economic inclusion may have for improving population health and reducing area-level health inequalities. At the same time, associations found between inclusive economy indicators and QALE were not always consistent with our hypotheses, highlighting the concept's complexity. One alternative approach would be to collapse information on all indicators into one single, averaging summary indicator, as it is often done with deprivation indices. However, one averaged summary indicator will likely fail to capture the diversity of inclusive economy indicators and the variation across different urban and rural areas. Here, more work is needed to refine and test how the inclusive economy can be captured.

## Conclusion

QALE is a comprehensive health expectancy measure and can be estimated for small areas in GB. QALE is conceptually close to the concept of QALYs, making it a comprehensive outcome measure for approaching area-level health inequalities, and evaluating past and future policies such as those seeking to establish more inclusive economies.

**Author affiliations**
[1]MRC/CSO Social and Public Health Sciences Unit, University of Glasgow, Glasgow, UK
[2]School of Geography, University of Leeds, Leeds, UK
[3]School of Medicine and Population Health, University of Sheffield, Sheffield, UK
[4]Manchester Institute of Education, The University of Manchester, Manchester, UK
[5]School of Social and Political Sciences, University of Glasgow, Glasgow, UK

**Acknowledgements** This research was conducted as part of the Systems Science in Public Health and Health Economics Research—SIPHER Consortium and we thank the whole team for valuable input and discussions that have informed this work. We would like to thank Emma Comrie and Katherine Smith for providing comments on an earlier draft of this paper, as well as Carl Schmertmann and Roland Rau for their strong commitment to open science. We are very grateful for the opportunity to work with the UK Household Longitudinal Study (Understanding Society). Understanding Society is an initiative funded by the Economic and Social Research Council and various Government Departments, with scientific leadership by the Institute for Social and Economic Research, University of Essex, and survey delivery by the National Centre for Social Research (NatCen) and Verian (formerly Kantar Public). The research data are distributed by the UK Data Service.

**Contributors** AH provided a first draft of the formal analysis and paper. AH, NL, HR and CH curated the underlying data. AH, NL, HR and CH contributed to the formal data analysis and validation of results. AH, NL, HR, CA, AB, DB, AC, CE, CH, SVK, GM, RS, AT and PM contributed to conceptualisation, study design, interpretation of findings, and the discussion of implications. All authors contributed to the first draft of the paper, the revision of the paper and have approved the final version. PM is the overall guarantor of this work.

**Funding** This work was supported by the UK Prevention Research Partnership (MR/S037578/2), which is funded by the British Heart Foundation, Cancer Research UK, Chief Scientist Office of the Scottish Government Health and Social Care Directorates, Engineering and Physical Sciences Research Council, Economic and Social Research Council, Health and Social Care Research and Development Division (Welsh Government), Medical Research Council, National Institute for Health Research, Natural Environment Research Council, Public Health Agency (Northern Ireland), The Health Foundation and Wellcome. In addition, this project received funding from the Medical Research Council (MC_UU_00022/5) as well as the Chief Scientist Office (SPHSU17 and SPHSU20). SVK acknowledges funding

from the European Research Council (949582), and the Medical Research Council (MC_UU_00022/2).

**Disclaimer** The funders had no role in the design of the study or in the collection, analysis, and interpretation of data and results.

**Competing interests** None declared.

**Patient and public involvement** Patients and/or the public were not involved in the design, or conduct, or reporting, or dissemination plans of this research.

**Patient consent for publication** Not applicable.

**Ethics approval** The University of Essex Ethics Committee has approved all data collection for the Understanding Society survey and the specific special licence linkage. We obtained a special licence version of Understanding Society (PN 228941). No additional ethical approval was necessary for other data sources or analyses.

**Provenance and peer review** Not commissioned; externally peer reviewed.

**Data availability statement** Data are available in a public, open access repository. We created a reproducibility pack for this paper: https://doi.org/10.17605/OSF.IO/26G98. The reproducibility pack contains detailed results as well as all code and data which we are able to share as open access material. We have published indicators of the inclusive economy as a separate data set with a detailed documentation: https://doi.org/10.17605/OSF.IO/VNSUR.

**ORCID iDs**
Andreas Höhn http://orcid.org/0000-0002-7170-1205
Nik Lomax http://orcid.org/0000-0001-9504-7570
Hugh Rice http://orcid.org/0000-0002-6895-8325
Colin Angus http://orcid.org/0000-0003-0529-4135
Alan Brennan http://orcid.org/0000-0002-1025-312X
Denise Brown http://orcid.org/0000-0002-5195-5312
Anne Cunningham http://orcid.org/0000-0003-0817-549X
Corinna Elsenbroich http://orcid.org/0000-0003-1153-4326
Ceri Hughes http://orcid.org/0000-0001-9495-8153
Srinivasa Vittal Katikireddi http://orcid.org/0000-0001-6593-9092
Gerry McCartney http://orcid.org/0000-0001-6341-3521
Rosie Seaman http://orcid.org/0000-0003-1400-4048
Aki Tsuchia http://orcid.org/0000-0003-4245-5399
Petra Meier http://orcid.org/0000-0001-5354-1933

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
