## [Reviewer comments · BMJ Open]

ARTICLE DETAILS

TITLE (PROVISIONAL)	Estimating Quality-Adjusted Life Expectancy (QALE) for Local Authorities in Great Britain and its Association with Indicators of the Inclusive Economy: A Cross-Sectional Study.
AUTHORS	Höhn, Andreas; Lomax, Nik; Rice, Hugh; Angus, Colin; Brennan, Alan; Brown, Denise; Cunningham, Anne; Elsenbroich, Corinna; Hughes, Ceri; Katikireddi, Srinivasa; McCartney, Gerry; Seaman, Rosie; Tsuchia, Aki; Meier, Petra

VERSION 1 – REVIEW

REVIEWER	Aburto, Jose Manuel The London School of Hygiene & Tropical Medicine
REVIEW RETURNED	04-Jul-2023

GENERAL COMMENTS	This paper presents a novel approach to study regional health inequalities by estimating Quality Adjusted Life Expectancy. The authors use state-of-the-art methods to estimate health status of the population at a geographical granular level that can be useful both to study inequalities in health and mortality, but also potentially to inform policy makers. I have only minor comments on this manuscript described below. Uncertainty is discussed thoroughly related to the sample from which health measures are derived, but I did not get whether uncertainty around the mortality estimates (from the TOPALS) is taken into account or included in the measures reported. How can this be integrated or why is not integrated is important to discuss. Fig 3, could the names be larger? They are difficult to read. The paper lacks a section of limitations. For example, the limitations of life expectancy as an indicator of longevity, especially in the context of the pandemic. It should be made explicit that it is not a projection or a forecast of any individual's lifespan and the assumptions should be explained in the methods section. Similarly, the Sullivan method does not account for heterogeneity and this should be explained. That is, when separating the nLx, essentially the survival function is being partitioned into different groups, but the survival function is from the population level mortality. It is likely that those in a non-healthy status have a different survival function. Overall this is a paper that I liked and enjoyed reading and I think is a contribution to the fields of health and demography.
--

REVIEWER	Mühlichen, Michael
-----------------	--------------------

	Federal Institute for Population Research
REVIEW RETURNED	13-Dec-2023

GENERAL COMMENTS	Thank you for this interesting article on QALE at a small spatial level in Great Britain, 2018-2020. While I do not think that a major revision is necessary, I have some minor concerns that should be addressed before publication:  1. The period spans 2018 to 2020. As we know from literature, the onset of the pandemic increased inequality in many contexts. Therefore, the inclusion of the year 2020 might distort the outcomes. The authors mentioned this limitation in the discussion but did not provide any information how this could be dealt with. Why not, for example, use the period 2017-2019 instead or analyse 2018/2019 and 2020 separately as a sensitivity analysis and describe the impact that the onset of the pandemic had in 2020 compared to previous years? 2. Why is Northern Ireland not included to "complete" the UK? 3. The term "Quality-adjusted" in QALE should be written consistently with a hyphen (-). (second paragraph.) In general, the manuscript should be handed to language editing before publication. While I think that the language is overall very good, there are some minor issues, e.g. regarding the setting of hyphens and commas. 4. In the third paragraph, the first two sentences start the same way. 5. QALY is mentioned as the gold standard but it is not explained why the authors use QALE instead. 6. The last para of the introduction mentions "inclusive economy" but does not explain what it means. 7. The packages to obtain the data seem to have flaws. While mid-year population counts are rounded to 100's, death counts for England and Wales seem to be censored for values lower than 5. For population, better data are available here: https://www.ons.gov.uk/peoplepopulationandcommunity/populationandmigration/populationestimates/datasets/populationestimatesforukenglandandwalescotlandandnorthernireland Uncensored death counts might be available here: https://www.ons.gov.uk/peoplepopulationandcommunity/birthsdeathsandmarriages/deaths/datalist?filter=datasets or here: https://www.nomisweb.co.uk/query/construct/summary.asp?mode=construct&version=0&dataset=161 8. Figure 3 shows a "distorted" map. While I understand the reason to take account of population density, many readers might find this illustration peculiar. I would recommend to also use "normal" shapefiles for LE and QALE on the LA level as supplementary material.
--

VERSION 1 – AUTHOR RESPONSE

Response to Reviewers

Reviewer: 1, Dr. Jose Manuel Aburto, The London School of Hygiene & Tropical Medicine

This paper presents a novel approach to study regional health inequalities by estimating Quality Adjusted Life Expectancy. The authors use state-of-the-art methods to estimate health status of the population at a geographical granular level that can be useful both to study inequalities in health and mortality, but also potentially to inform policy makers. I have only minor comments on this manuscript described below.

Uncertainty is discussed thoroughly related to the sample from which health measures are derived, but I did not get whether uncertainty around the mortality estimates (from the TOPALS) is taken into account or included in the measures reported. How can this be integrated or why is not integrated is important to discuss.

We fully agree. Approached via the Sullivan method, the uncertainty of health expectancies arises from its two component parts: the health and the mortality component. In our original submission, we have captured uncertainty arising from the health component only. In the revised version of our manuscript, we have now also incorporated uncertainty arising from the mortality component to provide a comprehensive picture. We have conceptualised the uncertainty of the mortality component to reflect the variance of the mortality rates which we have estimated via TOPALS models.

Please note that considering uncertainty arising from both components has increased the magnitude of previously presented uncertainty estimates (reflecting the health component only) by around 23% (unweighted median across all areas) We have reflected this change in our methodology throughout the manuscript, including our “Data & Methods” section:

“Approached via the Sullivan method, the uncertainty of health expectancies arises from the sum of its two component parts: the health and the mortality components [31]. We estimated the uncertainty of QALE point estimates considering these two components and as described by Jagger et al. 2014 [31]. While the variation of the health component reflects the uncertainty of the utility scores due to the number of survey participants, the variation of the mortality component reflects the uncertainty of estimated death rates and resulting death counts in the population which we obtained via TOPALS models. Across all LAs, the number of survey participants was substantially lower when compared to the actual population size of the respective LA. As a result, the overall amount of uncertainty is clearly dominated by the amount of uncertainty arising from the health component.”

Fig 3, could the names be larger? They are difficult to read.

We agree, this has been addressed. We have ensured that the font is increased, and that any overlap of labels is avoided. In our revised manuscript we now also provide a “classic” topographical map (Supplementary Figure 1).

The paper lacks a section of limitations. For example, the limitations of life expectancy as an indicator of longevity, especially in the context of the pandemic. It should be made explicit that it is not a projection, or a forecast of any individual’s lifespan and the assumptions should be explained in the methods section. Similarly, the Sullivan method does not account for heterogeneity, and this should be explained. That is, when separating the nLx, essentially the survival function is being partitioned into different groups, but the survival function is from the population level mortality. It is likely that those in a non-healthy status have a different survival function.

We fully agree with the reviewer and have incorporated the following changes to a, now, much more comprehensive “Strengths & Limitations” section:

1) Highlighting the limitations of life expectancy as a population health measure:

“As with life expectancy, and most measures of health expectancy such as HLE and DFLE, the estimation of QALE is based on a synthetic cohort approach. This means that our estimates provide timely cross-sectional snapshots - and should not be considered as precise forecasts of quality-adjusted life years that individuals are likely to experience. In the absence of shocks to population health, such as famines, epidemics, and wars, measures of life (and health) expectancy tend to underestimate the years of life (and years in good health) as societal and medical progress are not

accounted for [39]. On another note, recent period effects such as the consequences of austerity measures [40], the COVID-19 pandemic [41], and the subsequent cost-of-living crisis [42] highlight that deteriorations in population health can occur throughout the life course of individuals.”

2) Addressing the potential bias resulting from the usage of covid-year mortality data (also see comment of reviewer #2):

“The sensitivity of life and health expectancy to cross-sectional patterns of mortality and health means that our QALE estimates might have been impacted by the COVID-19 pandemic. The pandemic cannot have had an impact on the health component of our QALE estimates as the collection of survey data occurred throughout 2018 and 2019. However, the pandemic might have had an impact on the mortality component, which covers the years 2018 to 2020. As a sensitivity analysis, we therefore re-ran our analysis not considering mortality observed throughout the year 2020. A comparison of resulting differences in QALE is presented in **Supplementary Figure 2**. As expected, we found that excluding mortality of 2020 resulted in an (unweighted) increase of QALE of about 0.17 years among females and 0.28 years among males across all studied LAs.”

3) Pointing out that our approach does not reflect differences in mortality rates that likely exist between individuals of different health states:

“We obtained QALE estimates based on the Sullivan method. As a prevalence-based approach, the Sullivan method stipulates one universal mortality pattern, via the nLx parameter, for all individuals represented by a life table. This is a major methodological drawback, as it is likely that mortality will differ across individuals of different health status [43].”

Overall this is a paper that I liked and enjoyed reading and I think is a contribution to the fields of health and demography.

We would like to thank the reviewer for the positive and constructive feedback on our manuscript. We hope that our edits have addressed your comments satisfactorily.

Reviewer: 2, Dr. Michael Mühlichen, Federal Institute for Population Research

Thank you for this interesting article on QALE at a small spatial level in Great Britain, 2018-2020. While I do not think that a major revision is necessary, I have some minor concerns that should be addressed before publication:

1. The period spans 2018 to 2020. As we know from literature, the onset of the pandemic increased inequality in many contexts. Therefore, the inclusion of the year 2020 might distort the outcomes. The authors mentioned this limitation in the discussion but did not provide any information how this could be dealt with. Why not, for example, use the period 2017-2019 instead or analyse 2018/2019 and 2020 separately as a sensitivity analysis and describe the impact that the onset of the pandemic had in 2020 compared to previous years?

We fully agree with the reviewer. In line with the comment provided by reviewer #1, we now state in the discussion section:

“The sensitivity of life and health expectancy to cross-sectional patterns of mortality and health means that our QALE estimates might have been impacted by the COVID-19 pandemic. The pandemic cannot have had an impact on the health component of our QALE estimates as the collection of survey data occurred throughout 2018 and 2019. However, the pandemic might have had an impact on the mortality component, which covers the years 2018 to 2020. As a sensitivity analysis, we therefore re-ran our analysis not considering mortality observed throughout the year 2020. A comparison of resulting differences in QALE is presented in **Supplementary Figure 2**. As expected, we found that excluding mortality of 2020 resulted in an (unweighted) increase of QALE of about 0.17 years among females and 0.28 years among males across all studied LAs.”

We now also provide **Supplementary Figure 2**, which indicates the distribution of this change across all studied local authorities.

2. Why is Northern Ireland not included to "complete" the UK?

The primary focus of our consortium's work is Great Britain, and its 3 nations Scotland, England, and Wales. Due to this focus, we have not explored whether QALE can be estimated for local authorities in Northern Ireland (reflecting all of the United Kingdom). In addition, indicators captured in our inclusive economy indicator data set would have not allowed us to capture Northern Ireland. While the revised manuscript does not cover Northern Ireland, we might explore the option of obtaining results for Northern Ireland in upcoming future projects.

3. The term "Quality-adjusted" in QALE should be written consistently with a hyphen (-). (second paragraph.) In general, the manuscript should be handed to language editing before publication. While I think that the language is overall very good, there are some minor issues, e.g. regarding the setting of hyphens and commas.

Thank you very much for pointing this out. We have corrected this typo on page 5, and corrected a number of misplaced commas and semicolons.

4. In the third paragraph, the first two sentences start the same way.

We have corrected this in the introduction of our manuscript, now using "*These scores (...)*".

5. QALY is mentioned as the gold standard, but it is not explained why the authors use QALE instead.

Thank you for pointing this out. We have now clarified in the introduction:

“While QALYs are typically referred to as an outcome measure in longitudinal study designs, QALE reflects the application of utility scores to a synthetic life table cohort - analogously to the distinction enabled by other measures of health expectancy (e.g., disability-free life years (DFLY) vs. DFLE).”

6. The last para of the introduction mentions "inclusive economy" but does not explain what it means.

We have now clarified this in the introduction:

“Focusing on a range of different domains, the concept of the inclusive economy approaches the extent and nature of economic inclusion and participation. As economies are re-built following the COVID-19 pandemic, the concept of the inclusive economy might be more important for explaining population health outcomes and health inequalities than measures of economic growth or output [16].”

Please note that the underlying data set has now become publicly available.^[11]

7. The packages to obtain the data seem to have flaws. While mid-year population counts are rounded to 100's, death counts for England and Wales seem to be censored for values lower than 5. For population, better data are available

here: (1) <https://www.ons.gov.uk/peoplepopulationandcommunity/populationandmigration/populationestimates/datasets/populationestimatesforukenglandandwalescotlandandnorthernireland>

Uncensored death counts might be available

here: (2) <https://www.ons.gov.uk/peoplepopulationandcommunity/birthsdeathsandmarriages/deaths/datalist?filter=datasets>

or

here: (3) <https://www.nomisweb.co.uk/query/construct/summary.asp?mode=construct&version=0&dataset=161>

Thank you very much. To provide clarification, we have always used data on death counts in link (3). Your comment points towards the fact that ONS data can differ between routinely provided data (e.g., NOMIS) and bespoke data requests published retrospectively.

Due to the use of TOPALS models for modelling mortality rates against the general population standard, the overall impact of differences across ONS data sources is likely to be negligible. Our preference is to use NOMIS data for the following reasons:

- Open access: NOMIS data is available to everyone and does not rely on requests.
- Updated and sustainable: NOMIS data is updated once new estimates and methodologies are available while bespoke data requests will remain unchanged unless re-requested.
- Transparency: Purpose and details of the bespoke data requests are not always fully known.

We now acknowledge that other sources of ONS data exist and have edited the manuscript accordingly:

“We accessed data on mid-year population estimates and death counts for 5-year age groups via NOMIS. These routinely provided data included rounded mid-year population estimates (to 100s), while death counts below 5 were disclosed for LAs in England and Wales. NOMIS data can differ in comparison to bespoke data access requests available via the main ONS website. While bespoke data requests may contain more detailed information, they are not routinely updated - unless re-requested. To maximise reproducibility, transferability, and updateability of the underlying pipeline, we queried data in code-based format via NOMIS and its API. This reflects a sustainable and continuously updated source of data. Due to the use of TOPALS models for modelling mortality rates against the general population standard, the overall impact of differences across different data sources is likely to be negligible.”

Please note that at the time of manuscript submission (06/2023), the ONS has used census 2011 data to produce the most recent mid-year population estimates for all local authorities in England, Scotland, and Wales. In 11/2023, mid-year population estimates for local authorities in England and Wales were re-based and re-published using census 2021 data.^[2] Due to delays in the Scottish census, and the resulting delay in the publication of census results, re-based estimates are

not (yet) available for the local authority level in Scotland^[3]. This change in re-based estimates is now reflected in our manuscript and has led to minor changes in the exact magnitude of QALE estimates for local authorities. Please note that, still, all newly published estimates for England and Wales were derived via cohort component methods. We now state in our manuscript:

“To estimate life tables, we used official LA-level mid-year population estimates which are rolled forwards from the preceding census, using a cohort component model. The estimates based on the 2011 census have recently been revised to align with the 2021 census in England and Wales, but an equivalent adjustment has yet to be made in Scotland. Irrespective of adjustments, unrecorded international migration, and migration between LAs could have had an impact on the quality of mid-year population estimates, affecting the estimation of age- and sex-specific mortality rates. As the civil registration system in GB does not allow us to follow individuals’ place of residence between censuses, this numerator-denominator issue, alongside differences in data availability for the three GB nations, is likely to remain an unresolved challenge for periods between censuses [46].

8. Figure 3 shows a "distorted" map. While I understand the reason to take account of population density, many readers might find this illustration peculiar. I would recommend to also use "normal" shapefiles for LE and QALE on the LA level as supplementary material.

Thank you very much for the feedback. We have now also included the following topographical map as **Supplementary Figure 1** and link to this figure in our revised manuscript:

1

^[1] <https://osf.io/vnsur/>

^[2] [Rebasing of mid-year population estimates following Census 2021, England and Wales - Office for National Statistics \(ons.gov.uk\)](https://www.ons.gov.uk/news-releases/2022/03/2022-03-01-rebasing-of-mid-year-population-estimates-following-census-2021-england-and-wales)

^[3] <https://www.nomisweb.co.uk/articles/1368.aspx>

VERSION 2 – REVIEW

REVIEWER	Aburto, Jose Manuel The London School of Hygiene & Tropical Medicine
REVIEW RETURNED	08-Feb-2024

GENERAL COMMENTS	The authors have thoroughly addressed my concerns.
--

REVIEWER	Mühlichen, Michael Federal Institute for Population Research
REVIEW RETURNED	09-Feb-2024

GENERAL COMMENTS	Thank you very much for this improved version. All concerns have been addressed appropriately.
--